# Analysis on Isotropic and Anisotropic Samples of Polypropylene/Polyethyleneterephthalate Blend/Graphene Nanoplatelets Nanocomposites: Effects of a Rubbery Compatibilizer

**DOI:** 10.3390/polym16081092

**Published:** 2024-04-14

**Authors:** Vincenzo Titone, Marilena Baiamonte, Manuela Ceraulo, Luigi Botta, Francesco Paolo La Mantia

**Affiliations:** 1Department of Engineering, University of Palermo, V. le delle Scienze, 90128 Palermo, Italy; vincenzo.titone@unipa.it (V.T.); marilena.baiamonte@unipa.it (M.B.); manuela.ceraulo@unipa.it (M.C.); luigi.botta@unipa.it (L.B.); 2INSTM, Consortium on Materials Science and Technology, Via G. Giusti 9, 50125 Florence, Italy

**Keywords:** compatibilization, graphene nanoplatelets (GNPs), polyethylene terephthalate (PET), polymer blends, polypropylene (PP)

## Abstract

Over the past few years, polymer nanocomposites have garnered a significant amount of interest from both the scientific community and industry due to their remarkable versatility and wide range of potential uses in various fields, including automotive, electronics, medicine, textiles and environmental applications. In this regard, this study focuses on the influence of a compatibilizer rubber on a nanocomposite incorporating graphene nanoparticles (GNPs), with a polymer matrix based on a blend of polypropylene (PP) and polyethylene terephthalate (PET). This effect has been investigated on both isotropic samples and on anisotropic/spun fiber samples. The influence of the compatibilizer rubber on morphological, rheological and mechanical properties was analysed and discussed. Mechanical and morphological properties were evaluated on both isotropic samples obtained by compression moulding and melt-spun fibers. The addition of the rubbery compatibilizer increased the viscosity, improving interfacial adhesion, and the same effect was observed for the melt strength and breaking stretching ratios. Mechanical properties, including the elastic modulus, tensile strength and elongation at break, improved in both types of samples but more significantly in the fibers. These improvements were attributed to the orientation of the matrix, the formation of PET microfibrils, and the reduction in the size of graphene nanoparticles due to the action of the elongational flow. This reduction, facilitated by the elongation flow and the action of the compatibilizer, improved matrix–nanofiller adhesion due to the increased contact area between the two polymeric phases and between the filler and matrix. Finally, a transition from brittle to ductile behaviour was observed, particularly in the system with the compatibilizer, attributed to defect reduction and improved stress transmission.

## 1. Introduction

Polymer blends are a field of great interest for both scientific research and industry, as they offer a unique opportunity to combine the distinctive properties of different polymers [1,2]. However, this approach is not without its challenges, as a crucial problem emerges: the potential immiscibility between components [3]. In fact, this challenge requires innovative solutions, as the lack of miscibility between different polymers can lead to separate phases or regions of immiscibility, compromising the desired properties of the final material.

Among the many polymer blends investigated in the literature [4,5,6,7,8], a widely studied polymer blend concerns polypropylene (PP) and polyethylene terephthalate (PET) [9,10,11,12], two polymers with very different characteristics. Polypropylene offers remarkable impact and flexural strengths, while polyethylene terephthalate is distinguished by its excellent thermal properties and good chemical resistance [13]. The combination of these polymers promises materials with an optimal balance of strength, ductility and thermal stability [14]. However, the significant challenge of overcoming the inherent immiscibility between these two polymers stems from their different chemical structures and thermodynamic properties [3].

One strategy to address this problem is the introduction of compatibilizer agents. These agents play a crucial role in modifying intermolecular interactions between polymers, facilitating more homogenous mixing and ensuring a superior final performance.

In fact, many papers present in the literature show how the presence of compatibilizer agents in PP/PET blends promotes an effective interaction between the two surfaces [12,15,16,17,18,19]. Farikov et al. [15] observed a reduction in microfibril length attributed to the inhibitory effect of the compatibilizer on the coalescence process. Similarly, in our prior research [16], we noted a comparable trend: the microfibrils in ternary blends appeared to be shorter than those in binary blends. Mostovi et al. [17] demonstrated that microfibrils, when SEBS is present, exhibit lower aspect ratios, with a portion of SEBS adhered to the fracture surfaces, in contrast to the long, well-oriented microfibrils of PET without SEBS, as also corroborated by Park et al. [19] in PET/PP blended fibers utilizing a ternary ethylene ester acrylic–glycidyl methacrylate (EAG) copolymer. Of course, only a careful choice of compatibilizer agents can optimize the mechanical and thermal properties of these materials, making them more competitive.

Kraton™ polymers are styrene (ethylene-co-butylene)–styrene (SEBS) polymers functionalized with maleic anhydride (MA) grafted into a rubber intermediate block [20]. As previously reported in another paper [16], the presence of Kraton™ generates copolymers that act as a bridge between phases, enhancing the adhesion and properties of the polymer blend through a two-step reaction mechanism between the maleic ring and the terminal -OH group of PET.

In addition to the use of compatibilizer agents, another strategy to improve the properties of the polymer blends is the introduction of nanofillers that assume a key role in the improvement of these polymer blends. Indeed, as already reported in other works [21,22,23], graphene nanoplatelets (GNPs), with their unique properties, act as a structural reinforcement, bringing significant improvements to the mechanical and thermal properties of the composite material. In fact, in two distinct investigations, Inuwa et al. [21,22] examined the impact of incorporating styrene–ethylene–butylene–styrene grafted maleic anhydride (SEBS-g-MAH) and graphene nanoplatelets (GnPs) into a PP/PET blend. Their findings revealed that the nanoplatelets maintained their structural integrity and were uniformly dispersed within the polymer matrix, without experiencing an excessive exfoliation of the GnPs. Consequently, enhancements in mechanical properties were attributed to the inherent stiffness of the nanoplatelets and the efficient transfer of stress between the matrix and the filler. Furthermore, in a separate study, the researchers investigated the flammability characteristics of PP/PET/GnP nanocomposites, observing a notable enhancement in flammability parameters owing to the development of aligned, dense and well-organized carbon layers on the surface of the nanocomposites. Additionally, they noted an escalation in the effective thermal conductivity of the nanocomposites with increasing GnP loading.

For both polymer blends and nanopolymer blends, the final morphology is the key to obtain a material with the desired properties. To this aim, the elongational flow can play an important role. In fact, the flow involved in some processing operations, such as fiber spinning, film blowing and foaming, plays a crucial role in causing the preferential orientation of macromolecules along the flow direction. This flow-induced orientation significantly affects the deformation mechanism of the dispersed phase, thereby influencing the morphological evolution of blends and nanoblends, strongly affecting the final properties of the polymer system. Moreover, it has also evidenced that the elongational flow promotes the adhesion between the two, strongly affecting the final property phases of the polymer blends and between the polymer phase and the dispersed particles. This is due to the fact that the elongational flow is able to strongly decrease the dimension of the polymer phases and of nanofillers like GNP and nanoclay, increasing the area of the contact surface [24]. In our previous paper [16], it was demonstrated that the effect of the compatibilizer in a PP/PET blend that underwent elongational flow was negative as the increase in the mechanical properties was less than that observed for uncompatibilized blend. This behaviour was interpreted in terms of the reduced size of the initial PET droplet size in the ternary blend that gave rise to microfibrils of a shorter length than that of those formed during the spinning of the binary blend.

This study aims to evaluate the influence of a rubber compatibilizer and of the elongation flow on a GNP PP/PET nanocomposite. Our objective is to understand whether the presence of the compatibilizer affects the fibrillation efficiency of the dispersed PET phase. To this end, the mechanical and morphological properties of melt-spun fibers obtained with different hot drawing ratios were analysed and compared with isotropic samples. In addition, the rheological, morphological and mechanical properties of the isotropic samples obtained by compression moulding were evaluated.

## 2. Materials and Methods

### 2.1. Materials and Preparation

The main materials used in this work are as follows:Polypropylene (PP) with the trade name Moplen RP340H was purchased from LyondellBasell (LyondellBasell, Ferrara, Italy). It is a random polypropylene copolymer with a melt flow index (MFI) of 1.8 g/10 min (230 °C/2.16 kg) and a density of 0.90 g/cm) [25];Polyethylene terephthalate (PET) was obtained by grinding preforms for water bottles. Melt flow index (MFI) recorded at 275 °C under a load of 325 g (condition K) was 55 g/10 min [26];Kraton™ under the trade name Kraton™ FG1901 G was supplied from Kraton (Houston, TX, USA) as a dusted pellet. It is a clear, linear triblock copolymer based on styrene and ethylene/butylene with a polystyrene content of 30% [20];Graphene nanoplatelets (GNPs) under the trade name xGNP^®^, grade C, were supplied from XG Sciences Inc. (Lansing, MI, USA) with the following characteristics reported in a previous paper [23]: average diameter between 1 and 2 μm; an average thickness lower than 2 nm; and a specific surface area of about 750 m^2^/g.

The nanocomposites were prepared by melt mixer process in a Brabender mixer mod. PLE330 (Brabender, Duisburg, Germany) operating at 270 °C at 60 rpm for 5 min. Prior to blending, both PET and GNPs were dried in a vacuum oven at 120 °C overnight. Table 1 shows the codes and compositions of the nanocomposites investigated.

Mechanical properties were assessed on specimens obtained by compression moulding with a Carver laboratory hydraulic press (Carver, Wabash, IN, USA) at temperature of 270 °C and mould pressure of 300 psi for about 3 min. Fibers samples, ranging in diameter from 90 to 400 μm, were prepared at 270 °C utilizing the drawing module of a capillary viscometer (Rheologic 1000, CEAST, Turin, Italy).

Figure 1 illustrates the preparation and characterization of all the samples.

### 2.2. Characterization

Microscopic Raman examination was conducted at room temperature using a Renishaw InVia Raman microscope (Renishaw, Wotton-under-Edge, UK) equipped with a 532-nm Nd:YAG laser for excitation. Observations were carried out in the range of 3000–500 cm^−1^ with a spectral resolution of 0.5 cm^−1^.

Characterization in shear flow was conducted using an ARES G2 (TA Instruments, New Castle, DE, USA) rotational rheometer. Shear viscosity was assessed within the angular frequency range of 0.1 to 100 rad/s, employing a parallel plate geometry with a spacing of 1.5 mm, a diameter of 25 mm and a strain of 5% at a temperature of 270 °C. Additionally, flow curves at elevated shear rates were obtained using a capillary viscometer (Rheologic 1000, CEAST) featuring a capillary diameter of 1 mm and a length-to-diameter ratio of 40. In this study, the Bagley correction, which has to be applied to compensate for the pressure drop at the entrance of the capillary when its length-to-diameter ratio value is low, was omitted due to the high value of the length-to-diameter ratio. The Rabinowitsch correction was applied in order to calculate the true shear rate considering the non-Newtonian behaviour of the melt.

To evaluate the spinnability of these systems, on-isothermal elongational flow tests were performed employing the same instrument. A drawing module composed of a sequence of pulleys was used to grab the hot filament, which gradually cooled in the air, and transport it to an end pulley rotating at a consistent velocity. The force at break, known as the melt strength (MS), of the molten filament was directly measured. The breaking stretching ratio (BSR), denoted as the ratio of drawing speed at break to extrusion speed at the die, was determined, as described in the literature previously [27], by the following equation:(1)$$\mathrm{BSR}=\frac{V_{\mathrm{roll}}}{V_{p\;}\cdot \;\frac{D_{p}^{2}}{D_{c}^{2}}}$$
where V_roll_ represents the collecting speed, V_p_ signifies the capillary piston speed, D_p_ stands for the piston diameter and D_c_ denotes the capillary diameter.

Tensile tests were executed employing an Instron universal testing machine (Instron, mod. 3365, High Wycombe, UK) adhering to ASTM D638 [28]. The initial crosshead speed was set at 1 mm/min up to 3 min; afterwards, it was raised to 100 mm/min until failure of the specimen. The elastic modulus, E, tensile strength, TS, and elongation at break, EB, were determined with the mean value of eight measurements along with the corresponding standard deviation reported. Both isotropic sheets and oriented fibers were subject to characterization.

The drawing ratio, DR, of the fibers was computed, following the methodology detailed in a prior study [23], using the following formula:(2)$$\mathrm{DR}=\frac{D_{0}^{2}}{D_{F}^{2}}$$
where *D*_0_ represents the capillary diameter and *D_F_* signifies the diameter of the fibers.

The surface structure of the samples was analysed with a Quanta 200F scanning electron microscope (manufactured by a company in Hillsboro, OR, USA). Before SEM analysis, all samples were fractured in a cryogenic liquid and then coated with a gold layer to increase their conductivity. Image analysis was conducted using imaging software known as Image J, V. 1.53 which is freely available online.

## 3. Results and Discussion

Figure 2 shows the Raman spectra for the GNP, PPG and PPGK samples.

The presence of graphene nanoparticles (GNPs) within these nanocomposites was confirmed by our analysis of the Raman spectra. The Raman spectra of the GNPs showed three distinctive peaks at about 1326, 1590 and 2673 cm^−1^, associated with the D, G and 2D modes, respectively [29]. These peaks are also observed in the Raman spectra of the nanocomposites. The other peaks detected are related to the PP/PET blend.

Figure 3a–c show, respectively, the flow curves of the two nanocomposites measured with both the rotational rheometer and capillary viscometer (Figure 3a); the melt strength (MS) (Figure 3b); and the breaking stereo ratio (BSR) versus the shear rate (Figure 3c) for the two systems investigated.

Firstly, it can be seen that the curves obtained with the capillary viscometer do not coincide with the corresponding curves obtained with the rotational rheometer. This result is in agreement with those found in other similar studies [30,31,32] on heterogeneous, multiphase systems in which the Cox–Merz law is not respected. As regards to the significant increase in viscosity observed in the nanocomposite in the presence of the styrene (ethylene-co-butylene)–styrene copolymer grafted with maleic anhydride (SEBS-g-MA), this may be attributed to the better adhesion between the two phases. This improved adhesion, as previously reported in the literature [16], may be associated with the presence of the compatibilizer and the formation of copolymers which act as an adhesion promoter, reducing the interfacial tension and preventing coalescence. Consequently, the presence of Kraton™ provides more favourable support for optimizing the interaction between the components, contributing significantly to their flow behaviour.

The melt strength (MS) and breaking stretching ratio (BSR) are critical factors in some industrial processes involving elongational flow, particularly non-isothermal elongational flow, such as operations like spinning and film blowing. 

As expected, the melt strength (see Figure 3b) increases as the shear rate gradient increases, while the breaking stretching ratio decreases as it increases (see Figure 3c). Specifically, it is observed that the system with the compatibilizer exhibits higher values than the system without the compatibilizer, both in terms of the melt strength and breaking stretching ratios. Interestingly, this effect is mainly attributable to the increase in viscosity in the case of the melt strength, and, in the case of the breaking stretching ratio, the improvement in stretchability is due to the enhancement of adhesion at the interface of PP/PET.

Figure 4 shows typical stress–strain curves for the two nanocomposite systems obtained by compression moulding, while Table 2 summarizes the average values with their respective standard deviations.

It is evident from the stress–strain curve that material properties such as the elastic modulus, tensile strength and elongation at break significantly improve in the presence of the compatibilizer. More specifically, the most significant improvements are seen in the elongation at break, with a change from about 6.7% in the absence of the compatibilizer to about 16.7% in its presence. This result could also be attributed to the same reason as previously reported, i.e., the improved adhesion between the two phases. On the other hand, a relative increase of 11% in the elastic modulus and a 19% increase in the tensile strength are observed. 

Figure 5 shows a morphological analysis (SEM) of and the normal distribution curves for the isotropic samples.

The effect of the presence of the compatibilizer is well evident on both the PET phase and on the GNPs. Indeed, in the binary blend, the two phases are easily distinguished, the dispersed PET phase shows a large range of the dimensions of the particless and adhesion between the two phases is observed. In contrast, the two phases are almost indistinguable in the compatibilized blend. This is, of course, the result of the better adhesion between the PP and PET phases due to the formation of copolymers created by the reaction between the compatibilizer and the PET macromolecules. The GNPs are essentially concentrated in the PET phase and their dimension is strongly reduced in the compatibilized blend. Moreover, their distribution is also much more tight in the ternary blend, as is evident from the diameter distribution curves reported in Figure 5. The average diameter is about 0.2 um in the non-compatibilized blend, while it is only 0.1 um for the blend in the presence of the Kraton™. It is possible to interpret these data by considering the ternary blend has a larger viscosity and is able to undergo a lot of stress to break the GNPs.

The values for the elastic modulus, E, tensile strength, TS, and elongation at break, EB, of the fibers are shown in Figure 6a–c versus the drawing ratio (DR).

From Figure 6a, it can be seen that the elastic modulus increases with the hot drawing ratio. This is due to the orientation of the macromolecules, which is as a result of the applied elongation flow. However, the increase in the elastic modulus for both systems becomes negligible when the drawing ratio is more than about 60 and both curves tend to a plateau. On the other hand, the tensile strength and elongation at break (see Figure 6b,c) show a complex trend. Both the tensile strength and elongation at break first increase with the drawing ratio, reach a maximum and then start to decrease. This behaviour has already been noted in amorphous polymers, polymer blends and nanocomposites [23,30,31] and has been interpreted considering many factors. Firstly, the ordered morphology obtained during spinning, which aligns the macromolecules along the direction of flow, facilitates the sliding of the macromolecules during tensile testing. In addition, the microfibrils formed under the action of the non-isothermal elongational flow do not act as defects, unlike isotropic samples, during deformation [33]. Additionally, the large contact area between the matrix and PET fibrils reduces the impact of poor adhesion. However, once a certain drawing ratio (DR) is reached, in this case 60, the decrease in the elongation at break is probably due to the better orientation of the matrix, which makes the sample more brittle.

In Figure 7, the SEM micrographs of the fibers obtained at a DR = 60 and 100 for both samples are reported.

For both fibers at different drawing ratios, the PET phase exhibits a fibrillar morphology. However, in the non-compatibilized blend, these fibers are distinctly visible and separated from the matrix, whereas in the compatibilized blend, they are thoroughly embedded and firmly adherent. This improved adhesion undoubtedly contributes to the superior mechanical properties observed in the compatibilized blend compared to those of the binary blend. Furthermore, the graphene nanoplatelets remain smaller than those observed in isotropic samples. This reduction in size can be attributed to the elongational flow, which has the capability to further fragment the nanoplatelets into smaller dimensions. In fact, as can be seen in Figure 7a–d, as the drawing ratio increases, the particles undergo greater fragmentation, becoming smaller and thus less distinguishable. The box plot diagrams in Figure 8 clearly illustrate the different GNPs’ particle size distributions.

The isotropic PPG initially shows a wider size distribution (from 0.05 to 0.30 um), and the particle size of the GNPs (with an average value of 0.188 um) is larger than that of the isotropic PPGK (with an average value of 0.098 um). However, for fibers with a drawing ratio (DR) of 60, a reduction in the width of the size distribution is observed compared to the non-compatibilized isotropic sample (from 0.05 to 0.15). This reduction remains almost constant for both samples at the same DR; however, it is noted that the average particle size of the GNPs decreases for the compatibilized sample (0.08 um against 0.11 um). Similar results are found for both the PPG and PPGK with a DR of 100, showing a further decrease in the width of the size distribution and a decrease in the average particle size. This behaviour can be attributed to the action of the elongation flow, which further fragments the nanoparticles into smaller sizes with the increasing drawing ratio.

## 4. Conclusions

In this study, GNP nanocomposites, with and without the addition of a compatibilizer, were prepared by melt mixing with the aim of investigating the influence of a rubbery compatibilizer on the elongation flow of such systems. The most relevant results of this study are summarized in the radar chart in Figure 9 as follows.

In detail, the introduction of the compatibilizer in the nanocomposite blend results in an increase in melt viscosity compared to that of the corresponding sample without it, thus ensuring high melt strength values, which are essential for processing in some industrial processes, like spinning and film blowing. In addition, slight improvements in the BSR are shown due to better interfacial adhesion between the two phases. The mechanical properties of the isotropic sample, as can be seen from Figure 8, show slight improvements in the elastic modulus, tension and elongation at break. However, the most significant improvements, in terms of the elastic modulus, but especially the tensile strength and elongation at break, were observed on the melt-spun fibers. Indeed, the elongational flow acts on two different levels as follows:The chains of the PP matrix are oriented along the spinning line and the dispersed PET particles are elongated and oriented along the draw direction, reinforcing the polymer system and increasing the contact area between the two phases;The nanoplates are fragmented into smaller particles, increasing the contact area between the polymer and nanofiller.

Both of these actions give rise to a more ordered and homogeneous morphology with very high values at the interfaces. This morphology results in brittle to ductile behaviour and better values for the elastic modulus, tensile strength and elongation at break.

The compatibilizer and non-isothermal elongational flow affect the adhesion between the two phases of the blend and between the matrix and the graphene nanoplatelets; the fibrillation of the minor component of the blend; and the size reduction of the nanofiller. These actions result in the change of a more rigid polymer system into a more deformable polymer system, which is a very surprising and interesting result.

## Figures and Tables

**Figure 1 polymers-16-01092-f001:**
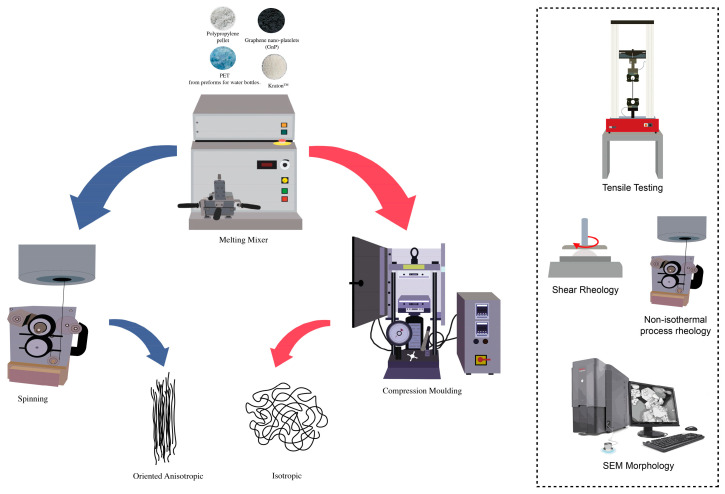
Image of the process of nanocomposites’ preparation and characterization.

**Figure 2 polymers-16-01092-f002:**
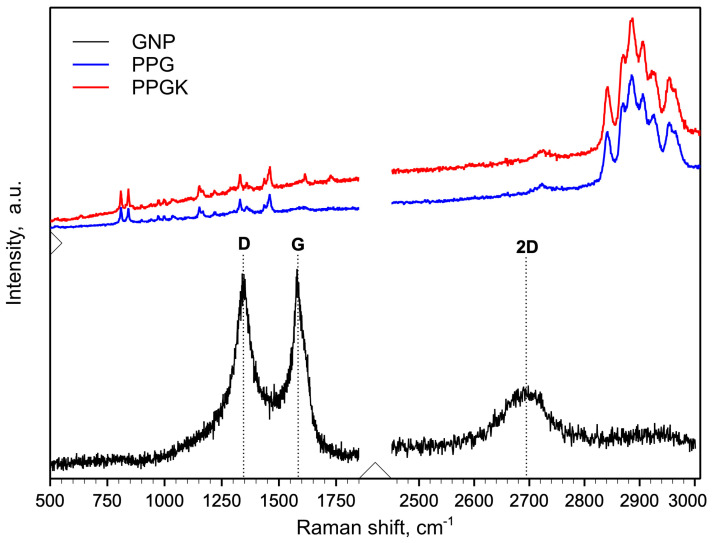
Raman spectra of GNP, PPG and PPGK samples.

**Figure 3 polymers-16-01092-f003:**
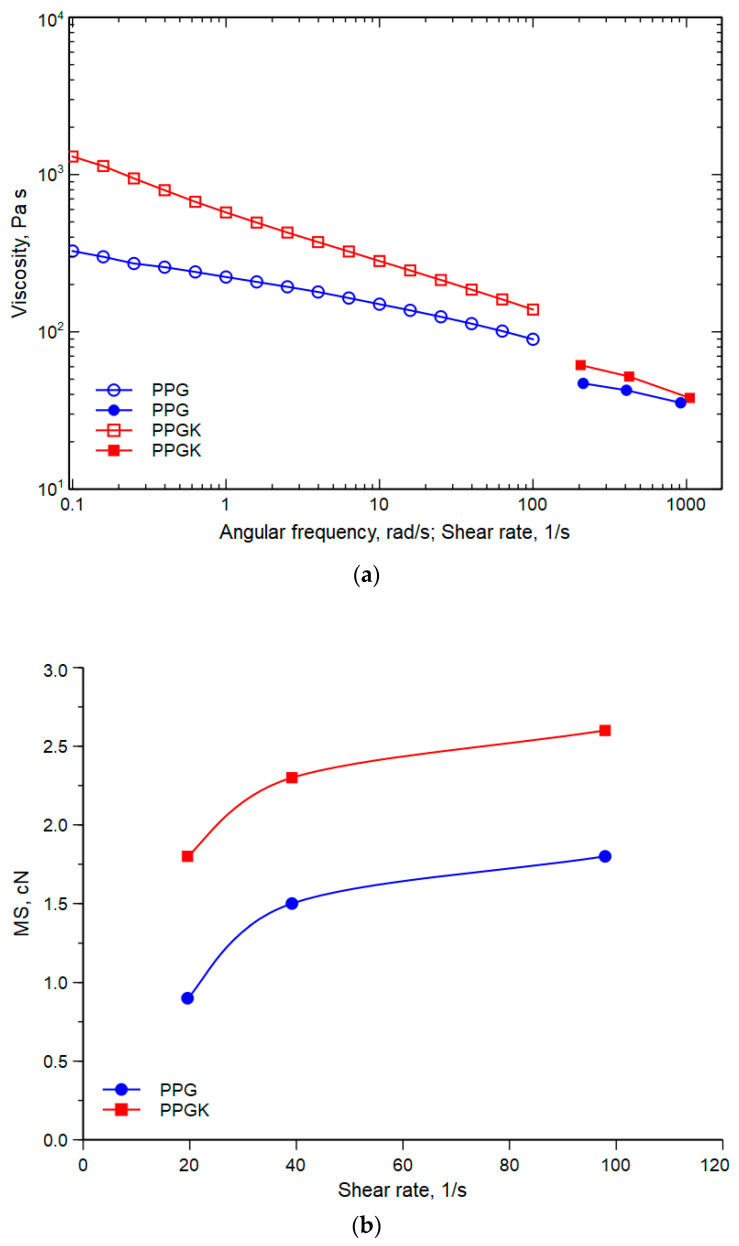
(**a**) Flow curves of nanocomposites: data represented with hollow shapes are acquired from the rotational rheometer and data represented by filled-in shapes are acquired from the capillary viscometer. (**b**) Melt strength (MS) versus shear rate of PPG and PPGK. (**c**) Breaking stretching ratio (BSR) versus shear rate of PPG and PPGK.

**Figure 4 polymers-16-01092-f004:**
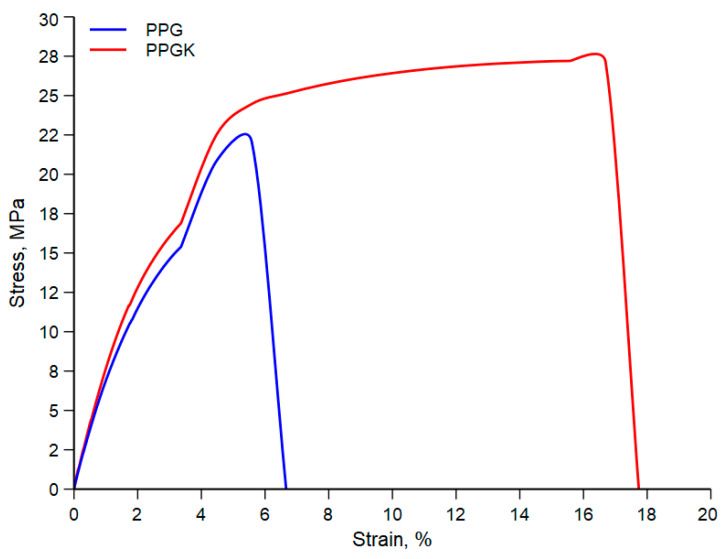
Typical stress–strain curves of isotropic sheets.

**Figure 5 polymers-16-01092-f005:**
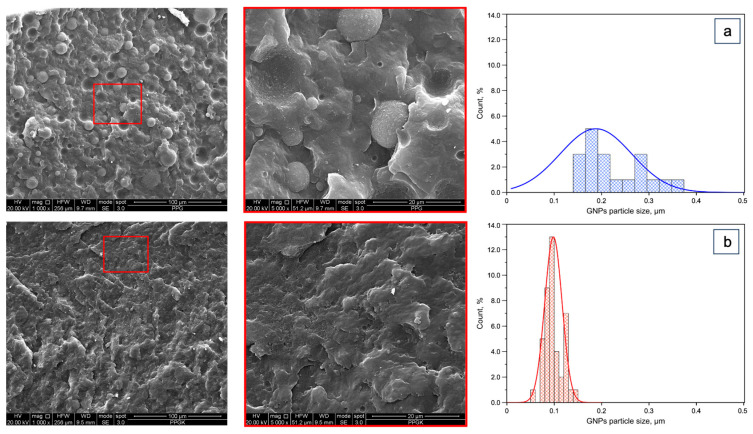
SEM micrographs and normal distribution curves of the isotropic sheets: (**a**) PPG; (**b**) PPGK.

**Figure 6 polymers-16-01092-f006:**
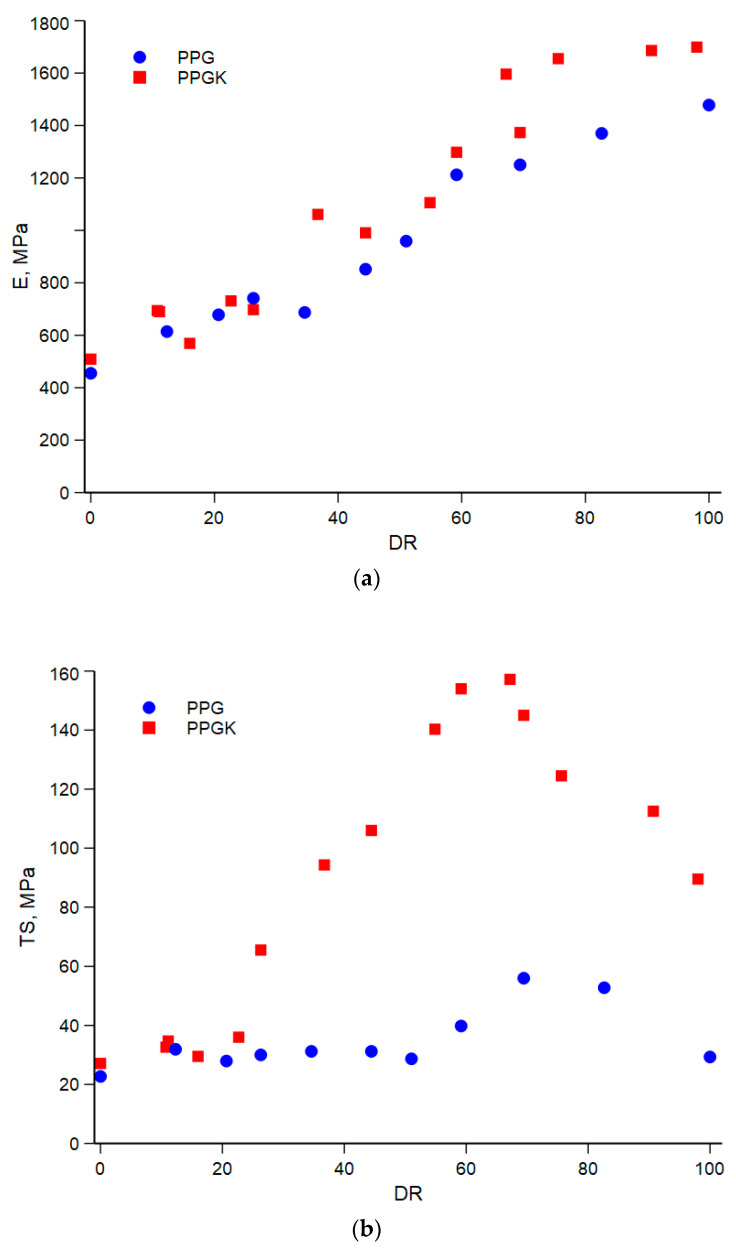
(**a**) Elastic modulus (E) versus drawing ratio (DR). (**b**) Tensile strength (TS) versus drawing ratio (DR). (**c**) Elongation at break (EB) versus drawing ratio (DR).

**Figure 7 polymers-16-01092-f007:**
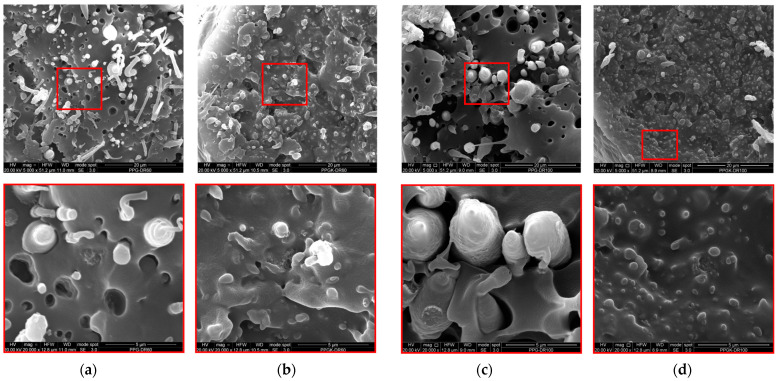
SEM micrographs of the anisotropic samples at DR = 60: (**a**) PPG and (**b**) PPGK; and DR = 100: (**c**) PPG and (**d**) PPGK.

**Figure 8 polymers-16-01092-f008:**
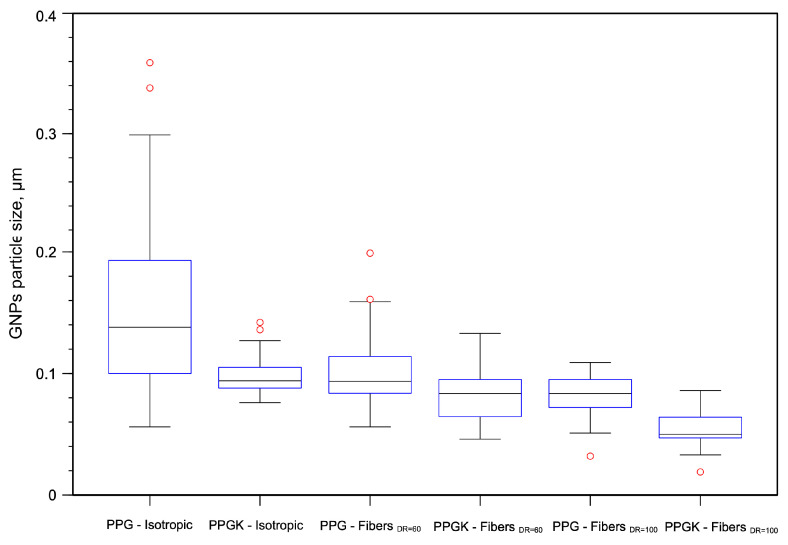
Boxplot of GNPs’ particle size distributions.

**Figure 9 polymers-16-01092-f009:**
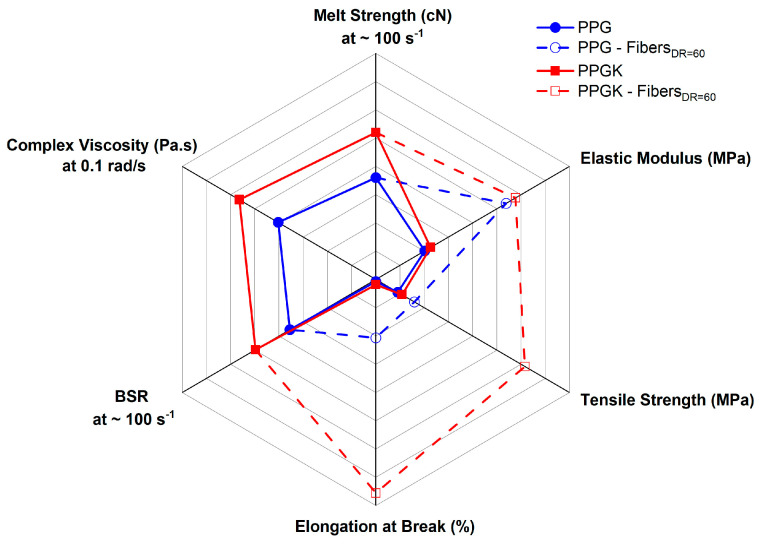
The properties of PPGK and PPG.

**Table 1 polymers-16-01092-t001:** Nanocomposites’ codes and compositions.

Sample Code	PP, wt%	PET, wt%	GNPs, wt%	Kraton™, wt%
PPG	75	25	2	-
PPGK	75	25	2	5

**Table 2 polymers-16-01092-t002:** Tensile properties of isotropic sheets.

Sample Code	E, MPa	TS, MPa	EB, %
PPG	455 ± 14	22.7 ± 0.9	6.7 ± 1.4
PPGK	509 ± 21	27.1 ± 1.4	16.7 ± 2.1

## Data Availability

Data are contained within the article.

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
