# Peer review of "Analysis on Isotropic and Anisotropic Samples of Polypropylene/Polyethyleneterephthalate Blend/Graphene Nanoplatelets Nanocomposites: Effects of a Rubbery Compatibilizer"

_polymers, 2024, doi:10.3390/polym16081092_

Round 1

Reviewer 1 Report

Comments and Suggestions for Authors

This study focuses on the influence of a compatibilizer rubber on the elongation flow of a nanocomposite incorporating graphene nanoparticles (GNPs).By using small-sized pet fibers and nanofillers, the contact area has been increased, thereby improving the transfer of stress from the PP matrix to the pet phase and nanosheets.But it still needs minor modifications before publication.

1. Please modify the language expression throughout the text, as the logic is confusing.

2. Please briefly describe the potential and innovation of nanocomposites in the abstract section.

3. Please provide a brief comparison of the differences in chemical structure and thermodynamic properties between PP and PET in the introduction section, in order to highlight the importance of overcoming their compatibility.

4. In section 3. Results and Discussion, please re explain the reason why both tensile strength and elongation at break increase first with the increase of tensile ratio, and then begin to decrease after reaching the maximum value. First, indicate the reason for the increase and then the reason for the decrease.

5. In the paragraph below Figure 10, "in the above two situations", the two situations were not clearly explained in the previous paragraph. Please elaborate on this in detail.

6. Please elaborate more on the beneficial outcomes of the two effects mentioned in the conclusion section.

Comments on the Quality of English Language

Extensive editing of English language required

Author Response

We thank the reviewer for their helpful comments. In attachment find a detailed point-by-point response to all comments

Reviewer 2 Report

Comments and Suggestions for Authors

In this manuscript, the author reported the Influence of a Rubbery Compatibilizer on GNPs Nanocomposites, The whole manuscript in relative good writing and organizing, some issues needed to be addressed.

(1) The polymer matrix is PP/PET, which should mentioned in the title.

(2) As mentioned in the title, “Analysis on Isotropic and Anisotropic Samples”, however, there was not preparation and description of Isotropic and Anisotropic Samples in the whole manuscript. In addition, how can you tell and why do you name Isotropic and Anisotropic Samples?

(3) Line 104, the MFI or molecular weight of PET should provided.

(4) Line 112, the unit of m2/g should revised.

(5) Line 114, at should changed to ,.

(6) Line 120, 300psi should 300 psi.

(7) Line 123 should changed to “Figure 1 illustrates the process of nanocomposites preparation and characterization.

(8) In the section of 3. Results and Discussion, the rheological and mechanical properties of the obtained nanocomposites should compared with the results in the literature, especially for other compatibilizer modified PP/PET composites.

Comments on the Quality of English Language

Extensive editing of English language required

Author Response

(The authors gave the same response as above.)

Reviewer 3 Report

Comments and Suggestions for Authors

The manuscript "Influence of a Rubbery Compatibilizer on GNPs Nanocomposites: Analysis on Isotropic and Anisotropic Samples" needs revision as there several parts in view of clarity and characterization that need to be improved.

1. In general abbreviations in title should be avoided please use full name of GNP. Also define abbreviations at place where they first appear. The title kind of too broad as PP/PET blends applied with GNP, so at least PP/PET blends should be mention.

2. The abstract far too long and confusing. Please shorten such and show the goal of this study and the most important results. In general abstract and conclusion should be partly coincide. Please modify such. The conclusion is very well presented. Additionally there some term not define (the reviewer suggest to explain such in the introduction) as example Cox-Merz rule. 

3. In the introduction there need be more example where either GNP as well other carbon based material added to the blends and which of them gave good improvements in strength and other rheological properties. The introduction should give some overviews of research made in similar way or similar approach increase the strength. As well the application of such composites are missing. Why as example are such blends as well carbon addition needed? How large is the improvement to pristine polymers? Please add more explanation why such research is needed.

4. The basic characterizations are missing such as FTIR and especially Raman to verify that GNP are added. Please include such. From reference 16 there are FTIR shown but if GNP added there need be some basic characterization (Raman at least should be added).

5. The flow of Figures in the manuscript using each measurement as own figure should be improved such as Figure 2-4 can be set as Figure 2a-c. It gives a better structure of the manuscript as well better organized. Aldo Figure 7-9 can be presented as Figure 5a-c. In general for research paper as a not written goal 6-8 Figures are ideal.

6. There need be some kind of Table of comparison to other works made in the field in view of strength and additives. Please add such before discussion as the readers need to understand  which level this research achieved in comparison to former one.

7. Minor corrections:

Page 2 line 61 "SEBS" as no definition shown what such is (it comes later but should be define where it first appear)

Page 2 line 74-79. That sentence starting with "Indeed, this type of flow..." need be reformulate as the sentence kind of confusing. The reviewer suggest make more than one sentence out of it.

Page 4, line 133 "Bagley correction" line 134 "Rabinowitsch correction" what are those and what do such mean. Ideal would be give a definition but at least references should be shown

Page 9 line 236 "comples" please correct that

Comments on the Quality of English Language

The English is fine just minor spell checking's required

Author Response

(The authors gave the same response as above.)

Round 2

Reviewer 2 Report

Comments and Suggestions for Authors

All of the issues mentioned were resolved in detail

Comments on the Quality of English Language

Moderate editing of English language required

Reviewer 3 Report

Comments and Suggestions for Authors

The authors made revision and responded to all suggestions as well added new experimental data. The manuscript now in acceptable form.